# Physicochemical Properties and Anticoagulant Activity of Purified Heteropolysaccharides from *Laminaria japonica*

**DOI:** 10.3390/molecules27093027

**Published:** 2022-05-08

**Authors:** Tingting Li, Haiqiong Ma, Hong Li, Hao Tang, Jinwen Huang, Shiying Wei, Qingxia Yuan, Xiaohuo Shi, Chenghai Gao, Shunli Mi, Longyan Zhao, Shengping Zhong, Yonghong Liu

**Affiliations:** 1Institute of Marine Drugs, Guangxi University of Chinese Medicine, Nanning 530200, China; li15578909861@126.com (T.L.); mhq18878839254@163.com (H.M.); hongli12212022@163.com (H.L.); yaoxuetanghao@outlook.com (H.T.); huangjinwen1127@163.com (J.H.); wsy980915@163.com (S.W.); qingxiayuan@163.com (Q.Y.); gaochh@gxtcmu.edu.cn (C.G.); mishunli@126.com (S.M.); 2Key Laboratory of Precise Synthesis of Functional Molecules of Zhejiang Province, School of Science, Westlake University, Hangzhou 310024, China; shixiaohuo@westlake.edu.cn

**Keywords:** *Laminaria japonica*, polysaccharides, physicochemical properties, anticoagulant activity

## Abstract

*Laminaria japonica* is widely consumed as a key food and medicine. Polysaccharides are one of the most plentiful constituents of this marine plant. In this study, several polysaccharide fractions with different charge numbers were obtained. Their physicochemical properties and anticoagulant activities were determined by chemical and instrumental methods. The chemical analysis showed that *Laminaria japonica* polysaccharides (LJPs) and the purified fractions LJP0, LJP04, LJP06, and LJP08 mainly consisted of mannose, glucuronic acid, galactose, and fucose in different mole ratios. LJP04 and LJP06 also contained minor amounts of xylose. The polysaccharide fractions eluted by higher concentration of NaCl solutions showed higher contents of uronic acid and sulfate group. Biological activity assays showed that LJPs LJP06 and LJP08 could obviously prolong the activated partial thromboplastin time (APTT), indicating that they had strong anticoagulant activity. Furthermore, we found that LJP06 exerted this activity by inhibiting intrinsic factor Xase with higher selectivity than other fractions, which may have negligible bleeding risk. The sulfate group may play an important role in the anticoagulant activity. In addition, the carboxyl group and surface morphology of these fractions may affect their anticoagulant activities. The results provide information for applications of *L. japonica* polysaccharides, especially LJP06 as anticoagulants in functional foods and therapeutic agents.

## 1. Introduction

*Laminaria japonica* belonging to the family Laminariaceae is a famous sea food and traditional medicine for its healthy and medicinal benefits. This marine plant is widely distributed and cultivated in many circumlittoral countries, such as China and Japan. Millions of tons of dried *L. japonica* are harvested every year in China [1,2]. For decades, numerous studies have demonstrated that *L. japonica* contains plenty of valuable nutrients and bioactive molecules, such as proteins, polypeptides, lipids, vitamins, and polysaccharides, which can affect human health status and treat some chronic diseases, such as metabolic syndrome, diabetes, obesity, and cardiovascular diseases [3]. Polysaccharides, as one of the main constituents of *L. japonica*, have attracted considerable attention due to their multiple bioactivities, such as immunomodulatory, anti-tumor, antioxidative, hypolipidemic, and anticoagulant effects [2,4]. It is well known that the physicochemical properties of polysaccharides, such as chain length, monosaccharide composition, charge number, and senior structure, are closely associated with their various biological activities [5,6]. However, according to previous studies [2,7,8,9,10,11], the physicochemical properties of *L. japonica* polysaccharides (LJPs) analyzed by different groups are significantly different, which may be due to the different extraction methods and source of plant materials. The physicochemical properties are clearly different, even when the LJPs have been obtained by the similar extraction and purification methods in previous studies [12,13]. Therefore, it is necessary to further elucidate the physicochemical properties of LJPs for their structural diversity and complexity limit the investigation of the structure–activity relationships and development of functional foods and drugs.

Thrombotic diseases are one of the leading causes of morbidity and mortality worldwide. The main option for the prevention and treatment of these diseases is drug therapy using anticoagulants [14]. Unfractionated heparin and low-molecular-weight heparin (LMWH) are still widely used by patients nowadays, although they have some side effects, such as serious bleeding and heparin-induced thrombocytopenia. In addition, they also have a risk of contamination by viruses such as prions, because they are mainly prepared from the mucosal tissues of bovine or porcine [15,16]. These problems have triggered the motivation for alternatives to heparin from non-animal sources, such as sulfated polysaccharides from plants [17]. Some studies have suggested that the sulfated polysaccharides from *L. japonica*, such as fucoidan, have significant anticoagulant activity, and polysaccharides with higher contents of sulfate groups exhibit higher anticoagulant activity [18,19]. In addition, it has been reported that the molecular weight and the molar ratio of fucose (Fuc) to galactose (Gal) of *L. japonica* polysaccharides may affect the anticoagulant activity [20]. However, how the LJPs exert the anticoagulant activity and whether other structural features influence the activity are still unknown. Our previous studies on sea cucumber polysaccharides indicated that molecular weight, carboxy groups, and sulfation patterns had obvious influence on their anticoagulant activity [5,21,22,23]. Therefore, the effects of physicochemical properties of LJPs on the anticoagulant activity and their action mechanisms still need to be further investigated.

In this study, we isolated several fractions from LJPs with different charge numbers using the anion-exchange column chromatography. Their physicochemical properties, including molecular weight, monosaccharide composition, and sulfate content, were analyzed. Their anticoagulant activities were evaluated by measuring their effects on activated partial thromboplastin time (APTT), thrombin time (TT), and prothrombin time (PT). The effect of these fractions on the intrinsic factor tenase (FXase) was further investigated to explore the underlying mechanisms. This study may provide helpful information for the development of LJPs as anticoagulants.

## 2. Results and Discussion

### 2.1. Extraction, Isolation, and Chemical Composition

LJPs were extracted with hot water and precipitated with ethanol. After removing proteins, pigments, and other small molecules, the yield of LJPs was 5.0% by dry weight of *L. japonica*. The contents of protein, Fuc, and sulfate groups were 1.3%, 24.3%, and 10.4%, respectively. These results suggested that acidic polysaccharides such as sulfated polysaccharides may exist in LJPs. LJPs were then separated by the DEAE-Sepharose FF column into four major polysaccharide fractions, named as LJP0, LJP04, LJP06, and LJP08, with yields of 35.5%, 6.6%, 8.9%, and 5.0% by dry weight of LJPs, respectively (Figure 1A). According to their HPLC profiles (Figure 1B), LJPs exhibited more than three peaks with molecular weights (Mw) between 5.5 and 1212.4 kDa. Compared with LJP0 and LJP04, LJP06 and LJP08 exhibited a relatively symmetrical peak, with Mw of 129.5 and 73.1 Da, respectively. The purified polysaccharide fractions did not contain proteins (Table 1).

The contents of sulfate and carboxy groups are essential to evaluate the charge distribution along the polysaccharide chain and the anticoagulant activity. The SO_3_^−^ to COO^−^ molar ratio is an important indicator determined in many studies on sulfated polysaccharides, which may also reflect the structural differences of various sulfated polysaccharides [24,25]. The conductimetric curves of LJPs and the purified fractions showed two intersections (Appendix A). The left inflexion point is the equivalence point of SO_3_^−^, and the right is that of COO^−^. The SO_3_^−^ to COO^−^ molar ratios of LJPs, LJP0, LJP04, and LJP06 were about 2.9–3.3 (Table 1), which are close to those of fucosylated glycosaminoglycan (FG) from different species of sea cucumbers [24,26]. However, the inflexion points in the conductivity titration curve of LJP08 were indistinguishable, which may be related to its chemical composition and structure.

The contents of sulfate group and chemical compositions of LJP0, LJP04, LJP06, and LJP08 demonstrated that contents of uronic acid and sulfate groups of these polysaccharide fractions increased with the increase in the concentration of NaCl elution solutions. The highest content of sulfate groups was observed in LJP08 among the purified fractions.

Figure 1C shows the monosaccharide compositions of LJPs and the purified fractions. LJP0 and LJP08 were mainly composed of mannose (Man), glucuronic acid (GlcA), Gal, and Fuc in molar ratios of 1:1.4:1.3:1.8and 1:2.1:1.4:2.0, respectively. LJP04 and LJP06 were mainly composed of Man, GlcA, Gal, xylose (Xyl), and Fuc in molar ratios of 1:1.8:0.4:0.3:1.1 and 1:2.3:0.7:0.2:1.1, respectively. These results suggested that polysaccharides derived from *L. japonica* were all heteropolysaccharides, which is consistent with previous reports [4,12]. LJP06 contained the most GlcA among the purified fractions. The monosaccharide compositions of LJPs determined by different groups are significantly different, which may be due to the different isolation methods and source of plant materials. For example, LJPs derived from *L. japonica* in Jeju, South Korea comprise thirteen monosaccharides [11]. Fuc, Gal, Man, GlcA, and mannuronic acid (ManA) are the predominant components. The study from Gao et al. [4] showed that the purified fractions have different monosaccharide compositions. LJ-A4 is composed of ManA, GlcA, Man, and arabinose (Ara); LJ-A6 comprises Fuc, Gal, ManA, GlcA, and Xyl; and LJ-A6 contains Fuc, Gal, ManA, and GlcA. Peng et al. [10] reported that a homogeneous polysaccharide (LJP12) is mainly composed of Ara, Xly, Man, Glc, and Gal. There are also the purified LJP fractions containing only Fuc and Gal [20]. Therefore, the monosaccharide compositions of LJPs and the purified fractions obtained in our study are obviously different from those reported by other groups. LJPs LJP0, LJP04, LJP06, and LJP08 do not contain ManA or Ara, and the contents of Fuc in these fractions are not as high as that of LJPs reported in some groups [20]. The main cause of structural difference of LJPs among different groups should be further elucidated in the future. According to our previous studies on FG from sea cucumbers, the structures of FG isolated from the same species of sea cucumber harvested from the same locations are almost no difference [5]. Therefore, we should focus on the extraction/separation methods, harvest seasons, and harvest locations to ensure the structural reproducibility of LJPs for drug development.

### 2.2. FT-IR and UV Spectra Analysis

Figure 1D shows the IR spectra of LJP0, LJP04, LJP06, and LJP08. In these spectra, the broad intense peak at around 3460 cm^−1^ and 1031 cm^−1^ can be assigned to the stretching vibration of O-H and C-O, respectively. The peak at around 2934 cm^−1^ was due to the stretching vibration of C-H. A strong peak at 1635 cm^−1^ was attributed to the asymmetric stretching vibration of C=O of GlcA. The peaks at around 1720 cm^−1^ and 1258 cm^−1^ were assigned to the valence vibration of C=O and C-O vibration of *O*-acetyl groups, respectively [27]. The signal at around 1422 cm^−1^ can be attributed to the stretching vibration of C-O within COOH. The peaks at 1245 cm^−1^ and 852/813 cm^−1^ were from the stretching vibration of S=O of sulfate and the bending vibration of C-O-S of sulfate in an axial position, respectively [22]. According to a previous study, the region around 810–855 cm^−1^ can be attributed to different positions of the sulfate groups in the sulfated polysaccharide chain [4,28]. The absorption at 850 cm^−1^ may be due to a sulfate group at the axial C-4 position, and the absorption around 820 cm^−1^ may be attributed to a sulfate group at equatorial C-2 position. LJP04 showed only a single absorption peak at 820 cm^−1^, which indicated the presence of sulfate groups at position C-2 of sugar residues. Other purified fractions, such as LJP0, LJP06, and LJP08, had bands at around both 820 cm^−1^ and 850 cm^−1^, indicating that they may contain some sugar residues sulfated at both the C-2 and C-4 positions. The ratios of these two sulfated positions among the purified fractions were different according to the absorption bands at around 820 cm^−1^ and 850 cm^−1^. According to the results of the chemical compositions in Table 1 and the IR spectra, we can find that the higher the content of sulfate groups in the purified polysaccharide fractions LJP04, LJP06, and LJP08, the stronger absorption at around 1245 cm^−1^. The contents of uronic acid and sulfate group of LJP0, LJP04, LJP06, and LJP08 increased with the increase in the concentration of NaCl elution solutions. LJP0 had a stronger absorption band at around 1245 cm^−1^ than LJP04, although they contained similar contents of sulfate groups. The absorption of the C-O vibration of O-acetyl groups at around 1245 cm^−1^ may influence the results. However, the main reasons should be further investigated. In summary, these differences in the characteristic absorption bands at 1700–800 cm^−1^ indicated that the purified polysaccharide fractions were structurally different.

The LJPs showed a peak at about 280 nm in the UV spectrum (Appendix A), indicating the presence of a little protein. However, no absorption at 280 or 260 nm in the UV spectra of the purified fractions such as LJP0, LJP04, LJP06, and LJP08 was observed, indicating that they did not contain protein and nucleic acids. These results are consistent with their chemical compositions.

### 2.3. ^1^H and ^13^C NMR Analysis

The ^1^H NMR spectra of LJPs, LJP0, LJP04, LJP06, and LJP08 are shown in Figure 2. In their ^1^H NMR spectra, the signals at 5.0–5.5 ppm and 4.4–4.7 ppm could be attributed to the anomeric protons of α and β configurations of sugar residues, respectively [29]. The broad and overlapping signals in the region of 3.3–4.3 ppm could be assigned to the ring protons of the sugar residues. The signals at 2.0–2.2 ppm were from the methyl protons of acetyl groups in the polysaccharides, indicating that some of monosaccharide residues were *O*-acetylated [27]. The signals at 1.2–1.3 ppm could be readily assigned to the CH_3_ protons of Fuc residues [30]. According to the spectra of LJP04, LJP06, and LJP08, few signals of impurities were found, indicating high purities of these purified fractions.

The ^13^C NMR spectra of LJPs, LJP0, LJP04, and LJP06 are shown in Figure 3. Signals of LJPs and LJP0 were very weak because of their high Mw. According to data from previous reports, the signals at 176−179 ppm could be attributed to the carbons of carbonyl groups in GlcA and acetyl groups [22,27,31], and the signals at about 96–111 ppm could be attributed to the anomeric carbons of sugar residues [30]. The chemical shifts at 70−88 ppm showing broad and overlapping peaks could be assigned to other carbons of glycosidic rings. The chemical shifts at δ 63 ppm may be due to the unsubstituted C-6 signals of Gal and Man. The up-field signals in the vicinity of 32/24 and 18 ppm could be attributable to the methyl carbons of acetyl group and Fuc residues, respectively [22]. These NMR signals of the purified fractions are consistent with the results of their chemical compositions. The detailed structures of these purified fractions need to be elucidated by their 2D NMR spectra in the future.

### 2.4. Morphological Characteristics

Scanning electron microscopy (SEM) is one of most commonly used qualitative method, which is also suitable for measuring the surface morphology of polysaccharides [29]. It has been reported that the spatial structures of polysaccharides may play an important role in their biological activity [32]. As shown in Figure 4, the SEM images of samples with magnification of 2000 showed that LJP0, LJP04, LJP06, and LJP08 exhibited significant variations in the surface morphology. LJP0 had a rough surface with many holes, and LJP04 presented a rough surface with few holes. LJP0 and LJP04 both had large wrinkles and rough structures on the surface, which may be due to their low content of sulfate group. A study from Liberman et al. showed that the sulfated polysaccharides of the red microalgae *Dixoniella grisea* and *Porphyridium aerugineum* containing low contents of sulfate groups had a porous fibrous structure with a rough amorphous surface, which is similar to that of LJP0 [33]. LJP06 showed a relatively even and smooth surface with several holes, and LJP08 was mainly composed of smooth slices with some crack. The molecular weight and chemical composition of sulfated polysaccharides may influence their surface morphologies according to some studies [34]. The surface morphologies of LJP06 and LJP08 also support the findings that the sulfated polysaccharides with high contents of sulfate group exhibit smooth structures with some rigid sheet-like fragments [35]. In this study, it seems that LJPs with a flat surface and in a denser texture may have higher bioactivities. This is consistent with the results of Cui et al., who reported that LJPs with a long-chain, uniform texture, flat surface, and denser crosslinking macromolecule exerted good immunomodulatory effects [11].

### 2.5. Anticoagulant Activity

According to the literature and our previous studies, heparin and LMWH have several side effects, such as serious bleeding, because they mainly target coagulation factors in the common pathway of the coagulation cascade that affects hemostasis [5,14]. The development of new drugs that inhibit components of the intrinsic coagulation pathway with a lower bleeding risk has become a research focus. APTT, PT and TT are important indicators of anticoagulant activity of compounds in vitro, and are widely used in screening anticoagulant active ingredients [36]. APTT is used to evaluate the effect of anticoagulant active ingredients on intrinsic coagulation pathway. The influence of compounds on the extrinsic coagulation pathway can be evaluated by PT through the detection of plasma coagulation time. TT can show the influence of samples on the common coagulation pathway. As shown in Table 2, the samples had no significant effect on PT in the concentration range of 200 μg/mL, indicating that they did not affect the extrinsic coagulation pathway. LJPs, LJP06 and LJP08 exhibited strong inhibitory activity against the intrinsic coagulation pathway. Their concentrations required to double the APTT of human standard plasma were 13~19 μg/mL. LJP0 and LJP04 showed a weak effect on APTT. Heparin, LMWH, LJPs, and LJP08 had strong effects on TT, indicating that they had a great influence on the common coagulation pathway. LJP0, LJP04, and LJP06 had little and weak effect on TT, respectively, indicating that they had a weak effect on the common coagulation pathway. Therefore, LJP06 mainly acted on the intrinsic coagulation pathway, which may have lower bleeding risk. Furthermore, these results suggested the higher the content of sulfate group in the purified fractions, the stronger the effects on APTT and TT. The molecular weights of these purified fractions may not have a decisive influence on their anticoagulant properties. According to previous studies, only certain chain length, such as three trisaccharide repeating units, may be sufficient for the strong anticoagulant activity of FG from sea cucumber [22].

These in vitro anticoagulant results encourage a much more detailed investigation on in vivo antithrombotic activity of the algae-sulfated polysaccharides to develop novel anticoagulant drugs. In addition, it has been reported that sulfated polysaccharides may be absorbed after oral administration and exert their anticoagulant activities [37,38]. This gives promise for further investigation on the health benefits of LJPs and the purified fractions, and to explore the feasibility of LJPs as functional foods with potential preventive applications.

A series of studies has demonstrated that inhibitors of the activated coagulation factors in the intrinsic pathway, such as factors FIXa, FXase, FXIa, and FXIIa, should effectively prevent thrombus formation with negligible bleeding risk [14,39,40]. The intrinsic FXase is an enzyme complex formed by FVIIIa, FIXa, Ca^2+^ and phospholipid, which is the final and rate-limiting enzyme complex in the intrinsic pathway [41]. Low-molecular-weight FG and its oligosaccharides from sea cucumber may serve as novel anticoagulants for their potent selective inhibition of the FXase without adverse effects [5]. LJPs, LJP06 and LJP08 had strong anticoagulant activity, shown by the activity to prolong APTT, which indicated the obvious effect of these polysaccharides on intrinsic coagulation pathway. Therefore, the effects of LJPs LJP06 and LJP08 on the intrinsic FXase activity were further determined and are presented in Figure 4. LJPs LJP06 and LJP08 potently inhibited the intrinsic FXase with EC_50_ about 25–30 ng/mL, which was close to that of LMWH. Given that LJPs and LJP08 had a great influence on the common coagulation pathway, LJP06 had a higher selectivity to inhibit FXase than LJPs and LJP08. LJP06 may have reduced side effects compared with heparin, LMWH, LJPs, and LJP08, which should be further confirmed in the future.

Previous studies have indicated that anticoagulant activity depends mostly on the molecular size, monosaccharide composition, content, and position of sulfate groups, and characteristic groups such as acetyl groups of polysaccharides [5,21,26]. In this study, based on the chemical compositions and anticoagulant activities of LJP0, LJP04, LJP06, and LJP08, we can determine that the higher the content of sulfate groups, the higher the anticoagulant activity. According to our previous study, carboxyl groups may also affect the anticoagulant activity of sulfated polysaccharides [21]. In this study, LJP06 and LJP08, containing relatively high contents of GlcA (Table 1 and Figure 3), had a higher inhibition to FXase than other purified fractions. The surface morphologies of these purified fractions may also affect their anticoagulant activities.

## 3. Materials and Methods

### 3.1. Materials and Chemicals

*L. japonica* was purchased from Xiapu County, Fujian Province, China. Deuterium oxide (D_2_O, 99.9% atom D), 3-Methyl-1-phenyl-2-pyrazolin-5-one (PMP), glucose (Glc), GlcA, Gal, galacturonic acid (GalA), Xyl, and rhamnose (Rha) were obtained from Sigma-Aldrich (St. Louis, MO, USA). Bovine serum albumin (BSA), trifluoroacetic acid (TFA), heparin, Fuc, ribose (Rib), arabinose (Ara), and Man were purchased from Aladdin Chemical Reagent Co., Ltd. (Shanghai, China). DEAE-Sepharose FF was from GE Healthcare (Uppsala, Sweden). CaCl_2_ solution (0.05 M), TT, PT, and APTT assay kits, and coagulation control plasma were obtained from TICO GmbH (Hamburg, Germany). Biophen FVIII: C kit was purchased from Hyphen Biomed (Neuvillesur-Oise, France). LMWH (0.4 mL × 4000 AXaIU) was from Sanofi-Aventis (Paris, France). Human factor VIII was obtained from Bayer Healthcare LLC (Leverkusen, Germany). All other chemicals, such as HCl, NaOH, and NaCl, were of analytical grade.

### 3.2. Extraction and Isolation of LJPs

The LJPs were extracted by the hot water extraction method, which is a classical and widely used method for polysaccharide extraction [1]. Briefly, 100 mg of dried *L. japonica* powder was added with 3 L of distilled water and extracted at 90 °C for 3 h. After extraction, the mixture was centrifuged at 4816× *g* for 15 min, and the supernatant was collected. The residue was added into 2 L of distilled water and extracted at 90 °C for 3 h again. The supernatant was added with triple volumes of ethanol, and kept overnight at 4 °C. After centrifugation at 4816× *g* for 15 min, the precipitates were dissolved in distilled water, and the protein was removed by an isoelectric point method [42]. Briefly, the polysaccharide solution was added with 6 M HCl to adjust the pH to 2–3, kept at 4 °C for 4 h, and centrifuged at 4816× *g* for 20 min. The obtained supernatant was adjusted to pH 7.0 with NaOH, and desalted using a dialysis bag with a 3500 Da molecular weight cut-off. The obtained polysaccharide solution was decolorized with a macroporous resin. Finally, the polysaccharide solution was condensed under reduced pressure and freeze-dried to obtain the LJPs.

The LJPs were isolated by an anion-exchange column loaded with DEAE-Sepharose FF resin. The column was gradient-eluted by 0, 0.2, 0.4, 0.6, 0.8, 1.0, 2.0 M NaCl solutions. The collected fractions were detected by the phenol–sulfuric acid method [43]. The sample was collected according to the absorbance value at 480 nm. Each polysaccharide fraction was desalted by the dialysis bag used previously. Finally, the samples were concentrated and lyophilized to obtain the purified polysaccharide samples named as LJP0, LJP04, LJP06, and LJP08.

### 3.3. Physicochemical Properties of LJPs and the Purified Fractions

The total carbohydrate contents of LJPs LJP0, LJP04, LJP06, and LJP08 were measured by the phenol–sulfuric method established by Dubois et al., using Glc as a standard [43]. Their protein contents were analyzed using the method reported by Bradford et al. [44]. The Fuc contents were analyzed using the cysteine–phenol–sulfuric acid method [45]. The sulfate contents were measured by a classical turbidimetric method [46]. The sulfate/carboxyl groups were determined by a conductometric method, as described in our previous study [26].

The molecular weights of LJPs LJP0, LJP04, LJP06, and LJP08 were determined by a high-performance gel permeation chromatography (HPGPC) performed on LC-2030C 3D HPLC apparatus (Shimadzu Corp., Kyoto, Japan) equipped with a Shodex OHpak SB-804 HQ column (7 µm, 8 × 300 mm) and a refractive index detector (RID) [47]. The chromatographic conditions were as follows: a constant flow rate of 0.5 mL/min was used on a 30 min isocratic elution at 0.1 M NaCl solution; column temperature was maintained at 35 °C; pullulan standards with molecular weights of 344.0, 107.0, 47.1, 21.1, and 9.6 kDa were used for the molecular weight estimation.

The monosaccharide compositions of LJPs LJP0, LJP04, LJP06, and LJP08 were analyzed on an Agilent ZORBAX Eclipse Plus C18 column (4.6 × 250 mm, 5 μm) and guard using a reverse-phase HPLC system equipped with a DAD detector, in accordance with our previous study. The column temperature was 30 °C. The mobile phase was acetonitrile and 20 mM ammonium acetate solution (17:83, *v*/*v*) with pH 6.7, and its flow rate was 1.0 mL/min. Before the HPLC analysis, the monosaccharide standards and polysaccharide samples were chemically transformed into the PMP derivatives after they were hydrolyzed by 4.0 M trifluoroacetic acid (TFA) at 120 °C for 2 h. After analysis, chromatographic peaks were manually integrated, and the monosaccharides were quantified by an external calibration curve, which was fitted with linear regression.

The IR spectroscopy of LJPs LJP0, LJP04, LJP06, and LJP08 was performed in the range of 4000–400 cm^−1^ using KBr pellets on a Nicolet iS50 Fourier-transform infrared spectroscopy spectrometer (Thermo Fisher Scientific, Waltham, MA, USA).

^1^H and ^1^^3^C NMR spectra were determined with a Bruker Avance spectrometer of 600 MHz equipped with a ^13^C/^1^H dual probe at 298.1 K. The dried LJPs, LJP0, LJP04, LJP06, and LJP08 were dissolved in deuterium oxide (D_2_O, 99.9% D) at a concentration of 10–20 mg/mL.

### 3.4. Anticoagulant Activity Assays

TT, APTT, and PT of LJPs LJP0, LJP04, LJP06, and LJP08 were measured on a coagulometer (TECO MC-2000, Hamburg, Germany.) using standard human plasma and TT/APTT/PT kits, as described in our previous work, with minor modifications [22]. Briefly, for analysis of TT, a 10 μL aliquot from sample solution or blank control solution (Tris-HCl buffer) was added into a test tube preheated at 37 °C, and 90 μL of standard human plasma was then added and incubated at 37 °C for 2 min. Finally, 50 μL of TT reagent preheated at 37 °C was added vigorously, and timing was started to record the coagulation time.

For assays of APTT, a 5 μL aliquot from sample solution or Tris-HCl buffer was added into a test tube preheated at 37 °C, and 45 μL of standard human plasma was then added and incubated at 37 °C for 2 min. Then, 50 μL of preheated APTT reagent at 37 °C was added, and incubated at 37 °C for 3 min. Finally, 50 μL of 0.02 M CaCl_2_ preheated at 37 °C was added vigorously, and timing was started to record the coagulation time.

For measurement of PT, a 5 μL aliquot from sample solution or Tris-HCl buffer was added into a test tube, and 45 μL of standard human plasma was added and incubated at 37 °C for 2 min. Finally, 100 μL of PT reagent preheated at 37 °C was added vigorously, and timing was started to record the coagulation time.

### 3.5. Inhibition of Intrinsic Factor Xase

Inhibition of the intrinsic factor Xase by LJPs LJP0, LJP04, LJP06, and LJP08 was measured in accordance with our previous studies using the BIOPHEN FVIII: C kit [21,22]. Briefly, a 30 μL aliquot from gradient concentration of the polysaccharide solutions and blank control solution (Tris-HCl buffer) were added to a 96-well cell culture plate, followed by 30 μL of FVIII solution (2 IU/mL), and 30 μL of R2 solution (60 nM IXa solution containing FIIa, Ca^2+^ and PC/PS). The 96-well plate was oscillated and incubated at 37 °C for 2 min. Additionally, then 30 μL of R1 solution mainly containing 50 nM FX solution and FIIa inhibitor was added, and the mixture was oscillated and incubated at 37 °C for 1 min. Finally, 30 μL of R3 solution containing 8.4 mM FXa-specific chromogenic substrate SXA-11 was preheated at 37 °C and added, and the absorbance value at 405 nm (OD_405_) was read continuously. The amount of FXa generation and the activity of FXase were represented by the change rate in absorbance at 405 nm (∆OD_405_/min).

### 3.6. Data Analysis

For the FXase assays, the EC_50_ was calculated by fitting the data to the equation: *B* = (EC_50_) ^n^/((EC_50_) ^n^ + [*I*] ^n^), where B is the percentage of remaining activity, n is the pseudo-Hill coefficient, EC^50^ is the concentration of sample that causes a 50% inhibition of FXase, [*I*] is the concentration of sample used as an inhibitor. All data for each group are given as the means ± SD. The data were analyzed using IBM SPSS Statistics version 26.0. Data were evaluated by one-way analysis of variance (ANOVA), followed by Duncan’s multiple-range test (DMRT). *P* values less than 0.05 were considered statistically significant.

## 4. Conclusions

LJPs were obtained by the widely used hot-water extraction method, and four purified fractions LJP0, LJP04, LJP06, and LJP08 were then isolated from LJPs. Results on the physicochemical properties of these purified fractions showed that they mainly consisted of Man, GlcA, Gal, and Fuc with different molar ratios, and contained different contents of sulfate groups. The purified fractions from LJPs were all acidic heteropolysaccharides. The LJPs, LJP06 and LJP08 exhibited strong anticoagulant activities by acting on the intrinsic coagulation pathway. Furthermore, we found that LJPs, LJP06 and LJP08 could potently inhibit the intrinsic FXase to exert their anticoagulant activities. However, LJPs, LJP08, heparin, and LMWH had a great influence on the common coagulation pathway that affects physiological hemostasis and results in bleeding. LJP06 has a relatively high yield, high homogeneity, and a weak effect on the common coagulation pathway, which may be developed as an advantageous anticoagulant. In addition, the anticoagulant activities of these purified fractions are related to the content of sulfate group, i.e., the higher content of sulfate group, the higher anticoagulant activity. The carboxyl group and surface morphology of these purified fractions may also affect their anticoagulant activities. These results are promising for further investigation of the feasibility of LJPs and the purified fractions as novel anticoagulants and functional foods.

## Figures and Tables

**Figure 1 molecules-27-03027-f001:**
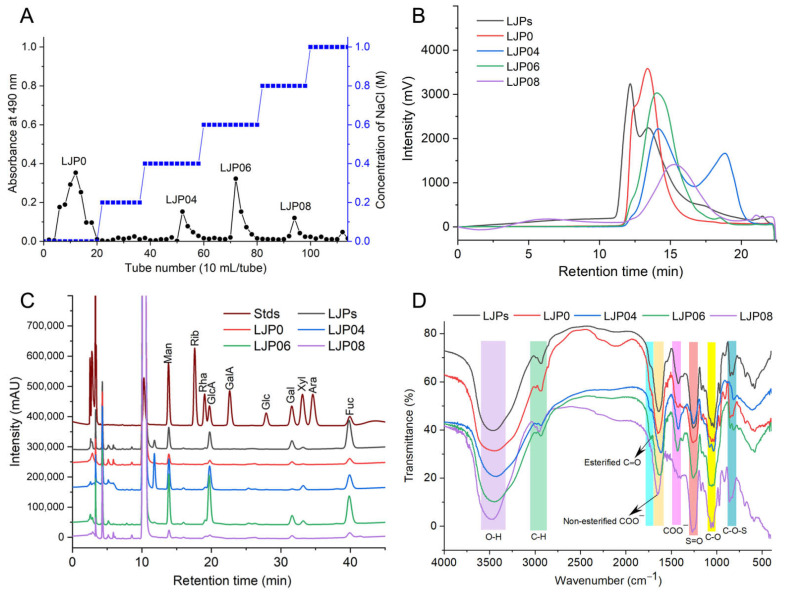
Preparation and physicochemical characteristics of LJPs and the purified fractions. Stepwise elution curve of LJPs (**A**), HPGPC profiles (**B**), monosaccharide compositions (**C**), and FT-IR spectra (**D**).

**Figure 2 molecules-27-03027-f002:**
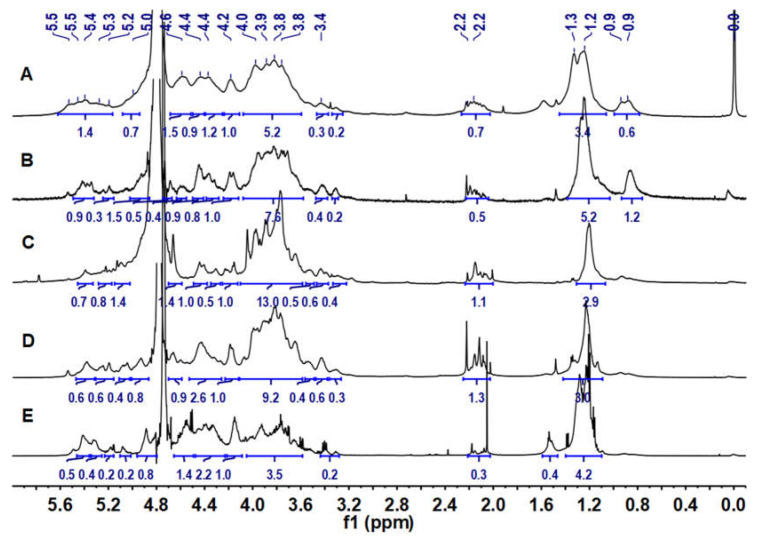
^1^H NMR spectra of LJPs (**A**), LJP0 (**B**), LJP04 (**C**), LJP06 (**D**), and LJP08 (**E**).

**Figure 3 molecules-27-03027-f003:**
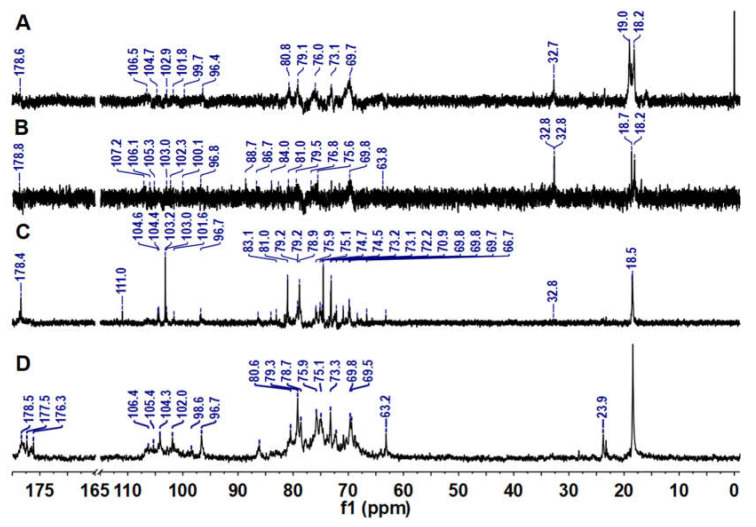
^13^C NMR spectra of LJPs (**A**), LJP0 (**B**), LJP04 (**C**), and LJP06 (**D**).

**Figure 4 molecules-27-03027-f004:**
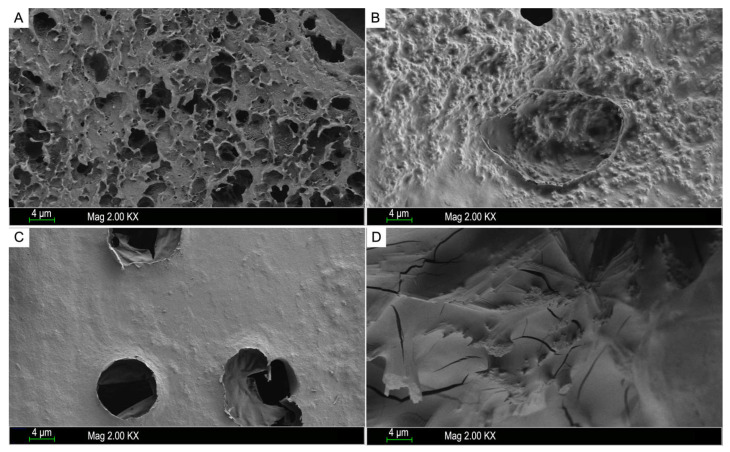
SEM images of the purified fractions LJP0 (**A**), LJP04 (**B**), LJP06 (**C**), and LJP08 (**D**). The magnification is 2 kX, and the scale bar is 4 μm.

**Table 1 molecules-27-03027-t001:** Chemical compositions and physicochemical properties of LJPs and the purified fractions.

Samples	Mw kDa	Protein (%)	Fuc (%)	OSO_3_^−^/COO^−^	Sulfate Group (%)	Chemical Composition (Molar Ratios)				
						Man	GlcA	Gal	Xyl	Fuc
LJPs	5.5~1212.4	1.3 ± 0.2	24.3 ± 2.5	2.94	10.4 ± 0.2	1	1.7	1.3	0.3	2.1
LJP0	1073.7, 22.7	0.2 ± 0.2	26.8 ± 0.3	2.95	3.5 ± 0.2	1	1.4	1.3	ND	1.8
LJP04	140.2, 7.8	−1.8 ± 0.6	12.0 ± 0.3	3.06	3.9 ± 0.6	1	1.8	0.4	0.3	1.1
LJP06	129.5	−0.9 ± 0.6	18.4 ± 0.3	3.25	9.5 ± 0.1	1	2.3	0.7	0.2	1.1
LJP08	73.1	−1.1 ± 1.0	24.6 ± 0.7	ND *	15.9 ± 0.1	1	2.1	1.4	ND	2.0

* ND: Not determined.

**Table 2 molecules-27-03027-t002:** Anticoagulant activity of LJPs LJP0, LJP04, LJP06, and LJP08 (*n* = 3).

Sample	APTT ^a^ (μg/mL)	TT ^a^ (μg/mL)	PT ^a^ (μg/mL)	Anti-FXase ^b^ (ng/mL)
Heparin	1.22 ± 0.19	0.43 ± 0.16	0.97 ± 0.08	8.30 ± 0.82
LMWH	3.96 ± 0.13	2.15 ± 0.09	>200	29.92 ± 3.50
LJPs	13.36 ± 0.82	12.48 ± 1.43	>200	27.6 ± 4.7
LJP0	>200	>200	>200	/ ^c^
LJP04	169.83 ± 23.01	>200	>200	/
LJP06	18.96 ± 4.33	65.77 ± 8.11	>200	24.9 ± 1.9
LJP08	17.90 ± 2.50	3.84 ± 0.80	>200	30.2 ± 4.3

^a^, The concentration required to double the APTT, TT, or PT of standard human plasma; ^b^, EC_50_, the concentration of each sample required to inhibit 50% of activity; ^c^, Not determined.

## Data Availability

The data that support the findings of this study are available from the corresponding author upon reasonable request.

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
