# Peer review of "Physicochemical Properties and Anticoagulant Activity of Purified Heteropolysaccharides from Laminaria japonica"

_molecules, 2022, doi:10.3390/molecules27093027_

Round 1

Reviewer 1 Report

The manuscript entitled “Physicochemical properties and anticoagulant activity of purified heteropolysaccharides from Laminaria japonica” was developed with the aim to evaluate the physicochemical properties and anticoagulant activities of several polysaccharide fractions, with different charge numbers, extracted from Laminaria japonica. The results provided information for the application of L. japonica polysaccharides (LJP) as anticoagulants in functional foods and therapeutic agents. The research design was developed appropriately and the methods were adequately described; however, results are not consistently discussed and this part of the text should be improved. In view of this, the manuscript is not acceptable for publication in this present form; some suggestions and corrections are presented below and should be considered by the authors:

Introduction:

Overall impression: LJP is the abbreviation of Laminaria japonica polysaccharides and sometimes is referred as singular, sometimes is referred as plural along the text. Please standardize this information. For example:

Lines 50-51: Please correct “Nevertheless, the physicochemical properties are obviously different, 50 even when the LJP are obtained by the similar extraction […].

Results and discussion:

FT-IR results: authors should explain better the peaks near to 1245 and 851 cm-1, respectively associated with stretching vibration of S=O of sulfate and the bending vibration of C-O-S of sulfate in an axial position. It is possible to observe that LJP 0 and LJP 06have similar behaviors; however, it is possible to observe that LJP04 present a different shift. Authors can correlate this information with the concentration of NaCl used to elute the anion-exchange column?

Morphological characteristics: discussion is mandatory. The morphological surface of other sulfated polysaccharides was already mentioned by the scientific literature and should increment this part of the text.

Anticoagulant activity: lines 204-205: Based on that they affirmed that LJP06 may have lower side effects than heparin, LMWH, LJP, and LJP08?

It is suggested that authors correlate the results obtained for anticoagulant activity and the results those reported by FT-IR, monosaccharidic composition, and NMR analyses.

Conclusion: how could LJP be applicated to functional foods? Please explain better in the results and discussion and provide a link for this information in the conclusions.

Author Response

Thank you for your comments concerning our manuscript entitled "Physicochemical properties and anticoagulant activity of purified heteropolysaccharides from Laminaria japonica". Those comments are all valuable and very helpful for revising and improving our paper. We have studied comments carefully and have made correction which we hope meet with approval. The main corrections in the paper and the responds to the comments are as following:

  1. Overall impression: LJP is the abbreviation of Laminaria japonica polysaccharides and sometimes is referred as singular, sometimes is referred as plural along the text. Please standardize this information. For example:

Lines 50-51: Please correct “Nevertheless, the physicochemical properties are obviously different, 50 even when the LJP are obtained by the similar extraction […].

Response: Thanks very much for your suggestion. We have revised the LJP to LJPs, and standardized this information in the new manuscript.

  1. FT-IR results: authors should explain better the peaks near to 1245 and 851 cm-1, respectively associated with stretching vibration of S=O of sulfate and the bending vibration of C-O-S of sulfate in an axial position. It is possible to observe that LJP 0 and LJP 06 have similar behaviors; however, it is possible to observe that LJP04 present a different shift. Authors can correlate this information with the concentration of NaCl used to elute the anion-exchange column?

Response: Thanks for your suggestion. We determined the IR spectra of the purified fractions again. The spectra of LJP0, LJP04, and LJP06 are the same as those determined previously. The spectrum of LJP08 is clearer than that measured previously. We have explained the peaks near to 1245 and 851 cm-1, respectively. According to previous study, the region around 810–855 cm−1 could be ascribed to different positions of the sulfate groups in the sulfated polysaccharide chain. The absorption at 850 cm−1 may be due to sulfate group at the axial C-4 position, and the absorption around 820 cm−1 may be ascribed to sulfate group at equatorial C-2 position. LJP04 showed only a single absorption peak at 820 cm−1, which indicated the presence of sulfate groups at position C-2 of sugar residues. Other purified fractions such as LJP0, LJP06, and LJP08 had the bands at around both 820 cm−1 and 850 cm−1, indicating that they may contain some sugar residues sulfated at both the C-2 and C-4 positions. The ratios of these two sulfated positions among the purified fractions were different according to the absorption bands at around 820 cm−1 and 850 cm−1. According to the results of chemical compositions in Table 1 and the IR spectra, we can find that the higher contents of sulfate group in the purified polysaccharide fractions LJP04, LJP06, and LJP08, the stronger absorption at around 1245 cm−1. LJP0 had a stronger absorption band at around 1245 cm−1 than LJP04, although they contained similar content of sulfate group. The absorption of C-O vibration of O-acetyl groups at around 1245 cm−1 may influence the results. Certainly, the main reason should be further investigated. The contents of sulfate group and chemical compositions of LJP0, LJP04, LJP06, and LJP08 demonstrated that contents of uronic acid and sulfate group of these polysaccharide fractions increased with the increase of the concentration of NaCl elution solutions. We have made these deep discussions as far as we can in the new manuscript.

  1. Morphological characteristics: discussion is mandatory. The morphological surface of other sulfated polysaccharides was already mentioned by the scientific literature and should increment this part of the text.

Response: Thanks very much for your suggestions. We have provided some discussion compared with the morphological surface of other sulfated polysaccharides in the new manuscript.

  1. Anticoagulant activity: lines 204-205: Based on that they affirmed that LJP06 may have lower side effects than heparin, LMWH, LJP, and LJP08?

Response: Thanks for your question. According to literature and our previous studies, heparin and LMWH have several side effects, such as serious bleeding, for they mainly target FIIa and FXa in the common pathway of the coagulation cascade (Lin, et al., Blood Rev. 2020, 39: 100615; Li, et al., Carbohydr Polym, 2021, 251: 117034; Zhao et al., PNAS, 2015, 112(27): 8284–8289.). In recent decades, anticoagulants that have emerged as alternatives to heparin-like drugs primarily target FIIa and FXa in the common pathway of the coagulation cascade but still exhibit adverse effects, particularly the risk of serious bleeding (Lin, et al., Blood Rev. 2020, 39: 100615). Components of the intrinsic coagulation pathway are promising targets for antithrombotic therapy because they are important for thrombosis but are not required for hemostasis. The development of new anticoagulant agents that inhibit components of the intrinsic pathway and that have a lower risk of causing bleeding has thus become a research focus (Zhao et al., PNAS, 2015, 112(27): 8284–8289; Li, et al., Carbohydr Polym, 2021, 251: 117034). A series of studies has demonstrated that inhibitors of the activated coagulation factors in the intrinsic pathway, such as factors FIXa, FXase, FXIa, and FXIIa, can effectively prevent thrombus formation with negligible bleeding risk (Zhao et al., PNAS, 2015, 112(27): 8284–8289.). It has been reported that FXase is the final and rate-limiting enzyme complex of the intrinsic coagulation pathway in the blood coagulation cascade; thus, it has been recognized as a potential target for discovering promising anticoagulants with lower bleeding risks (Mackman, Nature, 2008, 451 (7181): 914–918; Zhao et al., PNAS, 2015, 112(27): 8284–8289; Yin et al., J Biol Chem, 2018, 293(36): 14089-14099). We believe that LJP06 may have lower side effects such as serious bleeding than heparin, LMWH, LJP, and LJP08, because LJP06 has a higher selectivity to inhibit FXase in the intrinsic pathway. Certainly, this speculation needs to be further studied and confirmed in vivo. We have made some discussion in the new manuscript.

  1. It is suggested that authors correlate the results obtained for anticoagulant activity and the results those reported by FT-IR, monosaccharidic composition, and NMR analyses.

Response: Thanks for your suggestion. We have correlated the results of anticoagulant activity and those of molecular weight, FT-IR, monosaccharidic composition, and NMR analyses as far as we can in the revised manuscript.

  1. Conclusion: how could LJP be applicated to functional foods? Please explain better in the results and discussion and provide a link for this information in the conclusions.

Response: Thanks for your suggestion. We have explained the potential application of LJPs in functional foods better in the results and discussion and provide a link for this information in the conclusions. It has been reported that sulfated polysaccharides may be absorbed after oral administration and exert their anticoagulant activities (Fonseca and Mourão, Thromb. Haemost. 2006, 96, 822–829; Fonseca, et al., Thromb. Haemost. 2017, 117, 662–670). This gives promise for further investigation on the health benefits of LJPs and the purified fractions, and to explore the feasibility of LJPs as functional foods with potential preventive applications.

Reviewer 2 Report

Acceptance for publication of the article was based on:

The study presented by the authors focused on the phytochemical evaluation of a species of seaweed, known in the literature for its therapeutic virtues due to its content in proteins, polypeptides, lipids, vitamins and polysaccharides. The presence of sulphated polysaccharide – fucoidan, is notable for its anticoagulant effect, which is very well pointed out by the authors and is the aim of this study.

The introduction contains synthesized information on the species Laminaria japonica, regarding the correlation between the chemical composition and the induced therapeutic effects.

The species Laminaria japonica is used both for food and medicine, a very well-pointed aspect by the authors.

The species taken from the studio comes from a region of China, which is known for its large mass production of seaweed.

The authors set out in this experimental study to isolate several polyholoside fractions, to quantify them from a physico-chemical point of view, to determine the composition of the oxide chains and of course to highlight the anticoagulant effect.

All methods of analysis that contributed to this goal are described in detail.

The authors use modern methods of analysis specific to phytochemistry.

The isolation protocol of polysaccharide fractions is presented in detail. We started from specific property of these compounds, namely precipitation in alcohol. Extraction methods using solvent that do not generate toxins or toxins have been used. The cheapest solvent was used to extract these active ingredients, namely water. The polyholosides were precipitated in alcohol, and the precipitates were subjected to purification by resolubilization in hot water and reprecipitation. Several working methods have been used to isolate these fractions. Isolation yields of these polyholoside fractions were also mentioned.

Spectral analyses revealed the composition of each type of isolated fraction. A very important aspect in the research is represented by the SEM analysis of the fractions, finding a major differentiation between them. Evidence by attaching micrographs is greatly appreciated.

Investigating the anticoagulant effect in the acellular system is another important step in research. The effect induced on activated partial thromboplastin time (APTT), prothrombin time (PT) and thrombin time (TT) was followed. All tests were performed with a reference control, recognized for its anticoagulant action. The results obtained are presented in tables, easy to follow and interpret. The results obtained also allowed the issuance of very important hypotheses from a therapeutic point of view. The authors found that the higher the sulfate residue content in the structure of the polyholoside fractions, the greater the anticoagulant effect, with a direct action on the APTT and TT parameters. The effects induced on the coagulation parameters for each type of isolated and chemically characterized polyholoside fraction are described in detail.

To present the results of this study, the authors insert in the paper supporting tables with the experimental results obtained, the FT-IR, UV, 1H NMR, 13C NMR spectra are recorded.

The paper is accompanied by an additional material, which shows the UV spectrum of isolated polysaccharide fractions.

The conclusions are concise and point out the purpose and objectives of the proposed study.

The paper is scientifically supported by bibliographic indications, specific to the study question.

Author Response

Thanks for your comments.

Reviewer 3 Report

In this study, the authors reported the physicochemical properties and anticoagulant activity of polysaccharides extracted from laminaria japonica, the most famous seafood in China. The authors did plenty of works and the manuscript is well-written. However, there are a few major problems.

  1. polysaccharides from LJ have been well studied since probably 50 years ago, or even earlier, by a lot of researchers. The authors need to tell the readers what's new, either the analytical method or exciting results. However, many of the methods the authors used in the study are still as same as what i used 20 years ago. And all the results fell in the range of expectation and in line with previous studies. So this work has no novelty, to be frank. I suggest that the authors do some comparison of their results to some previous results. Find something interesting to the readers!
  2. in the part of IR and NMR analyses, the authors claims a few of annotations, which are correct from the scientific aspect. However, these kinds of description are so common in the analysis of fucoidan and can be found everywhere, even in a few of notebooks. Actually, there are a few of things that the authors can/should discuss. For example, in IR, the authors can discuss the relative ratio of peaks of SO3 and COO- and check if the IR results can match the results from chemical assay. in NMR, the authors may discuss the ratio of specific groups, e.g. COO-, to cross-check the content of GlcA. There are a few more things that should be interpret from IR and NMR.
  3.  Both fraction P0 and P04 should be further fractionated from GPC before further analysis.

some small problems:

  1. Line 105-111, the context discussing monosaccharide composition is inconsistent with the results in Table1

  2. when talking about centrifuge speed, dont use rpm. Please use g force (x g) 

  3. why did the author calculate SO3 to COO ratio? Why is the value of LJP08 not determined?

Author Response

  1. polysaccharides from LJ have been well studied since probably 50 years ago, or even earlier, by a lot of researchers. The authors need to tell the readers what's new, either the analytical method or exciting results. However, many of the methods the authors used in the study are still as same as what i used 20 years ago. And all the results fell in the range of expectation and in line with previous studies. So this work has no novelty, to be frank. I suggest that the authors do some comparison of their results to some previous results. Find something interesting to the readers!

Response: Thanks for your comments. There are many studies on the polysaccharides from L. japonica (LJPs). However, as stated in the Introduction of our manuscript, the physicochemical properties of L. japonica polysaccharides analyzed by different groups are significantly different, which may be due to the different extraction methods and source of plant materials. The physicochemical properties are obviously different, even when the LJPs are obtained by the similar extraction and purification methods in previous studies (Fang et al., Carbohydr. Polym. 2015, 134, 66–73; Chen et al., Int. J. Biol. Macromol. 2017, 105, 1421–1429.). Therefore, it is necessary to further elucidate the physicochemical properties of LJP for their structural diversity and complexity limit the investigation of the structure-activity relationships and development of functional foods and drugs. In our study, L. japonica was purchased from Xiapu County, Fujian Province, China, which is known as the “Hometown of kelp in China”. LJPs were separated by the DEAE-Sepharose FF column into four major polysaccharide fractions named as LJP0, LJP04, LJP06, and LJP08. We found that their chemical compositions and physicochemical properties were significantly different from those reported by some other groups. Although there are some studies on the anticoagulant activity in vivo and in vitro of the sulfated polysaccharides from L. japonica such as fucoidan, the underlying mechanisms and the relationship between structure and anticoagulant activity are still unknown. Especially, which coagulation factors in the pathway of the coagulation cascade targeted by LJPs are still unknown. In our study, we found that LJP06 mainly acted on the intrinsic coagulation pathway, and potently inhibited the intrinsic FXase to exert its anticoagulant activity, which may have lower bleeding risk. Therefore, our study has novelty and may provide some information for investigation and development of LJPs as anticoagulants. We have made some comparison of our results to some previous results as far as we can.

  1. in the part of IR and NMR analyses, the authors claims a few of annotations, which are correct from the scientific aspect. However, these kinds of description are so common in the analysis of fucoidan and can be found everywhere, even in a few of notebooks. Actually, there are a few of things that the authors can/should discuss. For example, in IR, the authors can discuss the relative ratio of peaks of SO3 and COO- and check if the IR results can match the results from chemical assay. in NMR, the authors may discuss the ratio of specific groups, e.g. COO-, to cross-check the content of GlcA. There are a few more things that should be interpret from IR and NMR.

Response: Thanks for your suggestion. IR is generally used to qualitatively analyze the main structural information of polysaccharides. Both the C-O vibration of O-acetyl groups and the stretching vibration of S=O of sulfate are at around 1245 cm−1, and different sulfate group location in the polysaccharide chain has different absorption at 810-855 cm−1. Hence, the relative ratio of peaks of SO3- and COO- cannot be calculated by the IR spectra. What’s more, since the signals on COO- cannot be observed in 1H NMR, the ratio of specific groups such as COO- cannot be calculated by the 1D NMR spectra. We have provided more discussions on this part in the new manuscript as far as we can.

  1. Both fraction P0 and P04 should be further fractionated from GPC before further analysis.

Response: Thanks for your suggestion. If we found the favorable bioactivities of LJP0 and LJP04, we would further fractionate them. In the present study, LJP0, and LJP04 had little anticoagulant activity. Hence we did not further fractionate them. At present, we focus on the studies of LJP06 for it has relatively high yield and highly selective inhibition to the FXase.

  1. Line 105-111, the context discussing monosaccharide composition is inconsistent with the results in Table1

Response: Thanks for your suggestion. The data of monosaccharide compositions in Table 1 are calculated from those given in the text. Now, we have revised the data in the text to bring into correspondence with the data in the Table 1 in the new manuscript.

  1. when talking about centrifuge speed, dont use rpm. Please use g force (x g)

Response: We have changed the unit of centrifuge speed "rpm" to "g force (x g) " in the new manuscript.

  1. why did the author calculate SO3 to COO ratio? Why is the value of LJP08 not determined?

Response: Thank you for your question. The contents of sulfate and uronic groups are essential to evaluate the charge distribution along the polyelectrolyte chain. This indicator is determined in many studies on the sulfated polysaccharides, which may also reflect the structural differences of different sulfated polysaccharides (Luo, et al., Mar. Drugs 2013, 11, 399-417; Shi, et al., Mar. Drugs 2021, 19, 162.). We have explained these in the new manuscript. The molar ratio of -OSO3/-COO of LJP08 cannot be calculated by its conductivity titration curve for its inflexion points are indistinguishable, which may be related to its structure. We have provided the conductivity titration curves of LJPs and its purified fractions in the Supplementary materials and discussed them in the revised manuscript clearly.

Round 2

Reviewer 1 Report

The authors made a proper revision of the original paper and tried to respond to all of the questions elaborated by the reviewer. However, it is possible to find some details that should be corrected before acceptance.

The authors described this text in the response to the reviewer´s and, in my apnion, this should be added to the discussion of FTIR: "The contents of the sulfate group and chemical compositions of LJP0, LJP04, LJP06, and LJP08 demonstrated that the contents of uronic acid and sulfate group of these polysaccharide fractions increased with the increase of the concentration of NaCl elution solutions."

Line 263: please correct: "The intrinsic FXase is an enzyme complex formed by factor VIIIa (FVIIIa), factor IXa (FIXa) [...]".

Lines 282-284: that phrase is confusing; please revise this part in order to improve the understanding of the context.

Author Response

Thank you for your comments, and those comments are all valuable and very helpful for revising and improving our paper. The main corrections in the paper and the responds to the comments are as following:

The authors described this text in the response to the reviewer´s and, in my apnion, this should be added to the discussion of FTIR: "The contents of the sulfate group and chemical compositions of LJP0, LJP04, LJP06, and LJP08 demonstrated that the contents of uronic acid and sulfate group of these polysaccharide fractions increased with the increase of the concentration of NaCl elution solutions."

Response: Thanks for your suggestion. We have added this content to the discussion of FTIR.

Line 263: please correct: "The intrinsic FXase is an enzyme complex formed by factor VIIIa (FVIIIa), factor IXa (FIXa) [...]".

Response: Thanks for your suggestion. The sentence is correct (Lin, et al., Blood Rev. 2020, 39: 100615; Zhao et al., PNAS, 2015, 112(27): 8284–8289). We have revised the “factor VIIIa (FVIIIa), factor IXa (FIXa)” to the “FVIIIa, FIXa”.

Lines 282-284: that phrase is confusing; please revise this part in order to improve the understanding of the context.

Response: Thanks for your suggestion. We have put this sentence to the part “2.3. 1H and 13C NMR analysis” and revised it.

Reviewer 3 Report

All of my previous comments have been successfully addressed.

I have another small suggestion. Marine polysaccharides for drug candidate are interesting. However, the structure of polysaccharides (especially fucoidan) from the same brown algae may be different, due to different extraction/separation methods, harvest seasons, harvest locations, ect. Sometimes, the harvest seasons and harvest locations may contribute more to the structural heterogenicity than the extraction method do. So, the major challenge is that after a group find a fraction with good bioactivity, how can the other groups get the same fraction, or say how can we ensure the structural reproducibility for a drug (candidate)?

I expect that the authors can point out this significance and/or give a small discussion.

Author Response

Thank you for your comments, and those comments are all valuable and very helpful for revising and improving our paper. The main corrections in the paper and the responds to the comments are as following:

I have another small suggestion. Marine polysaccharides for drug candidate are interesting. However, the structure of polysaccharides (especially fucoidan) from the same brown algae may be different, due to different extraction/separation methods, harvest seasons, harvest locations, ect. Sometimes, the harvest seasons and harvest locations may contribute more to the structural heterogenicity than the extraction method do. So, the major challenge is that after a group find a fraction with good bioactivity, how can the other groups get the same fraction, or say how can we ensure the structural reproducibility for a drug (candidate)?

I expect that the authors can point out this significance and/or give a small discussion.

Response: Thanks very much for your comments and suggestions. The structure of LJPs determined by different groups are significantly different, which may be due to different extraction/separation methods, harvest seasons, harvest locations, ect. However, the real reasons are still unknown. In the future, we can further investigate the main reasons. According to our previous studies on the fucosylated glycosaminoglycans (FG) from different species of sea cucumbers, the structures of FG isolated by similar extraction method from the same species of sea cucumber that was harvested from the same locations are almost no difference (Li, et al., Carbohydr Polym, 2021, 251: 117034). We have discussed these in the section “2.1. Extraction, isolation and chemical composition”.